# Origins of the hydrothermal dolomites in Middle Permian, Sichuan Basin (SW China): Implication for the relationship with the Emeishan Large Igneous Province

Haofu Zheng[1], Jiajun He[1], Xiong Duan[2]*, Bo Liu[3], Anna Travé Herrero[4], Juan Diego Martín-Martín[4], Enrique Gómez Rivas[4]

1 College of River and Ocean Engineering, Chongqing Jiaotong University, Chongqing, China, 2 School of Geographical Sciences, Sichuan Provincial Engineering Laboratory of Monitoring and Control for Soil Erosion in Dry Valley, China West Normal University, Nanchong, China, 3 School of Earth and Space Sciences, Peking University, Beijing, China, 4 Departament de Geoquímica, Petrologia i Prospecció Geològica, Universitat de Barcelona, Barcelona, Spain

* duanxiong00@163.com

## Abstract

The Middle Permian dolomites in the Sichuan Basin (SCB) of China include several replacive phases and pore-filling saddle dolomite (Rd3 and Sd). This study investigates the origin of the main hydrothermal phases (Rd3 and Sd), which are composed of medium to coarse, non-planar crystals and are heterogeneously distributed, with abundance decreasing sequentially from the southwestern (SW) to the central and northwestern (NW) SCB. A comprehensive petrographic and geochemical analysis reveals a systematic spatial gradient across the basin: fluid inclusion temperatures, salinities, dolomite ordering degrees, and calculated $\delta^{18}O_{fluid}$ values all increase from the NW to the SW. This trend reflects a transition from a distal, rock-buffered diagenetic system in the NW, dominated by heated Permian seawater with lower temperatures and less evolved fluid chemistry, to a proximal, fluid-buffered system in the SW. The SW system was controlled by the advective influx of high-temperature, high-salinity basinal brines with a strong crustal signature (e.g., highly radiogenic $^{87}Sr/^{86}Sr$ ratios and enriched $\delta^{18}O_{fluid}$ values), which ascended along basement faults. This large-scale hydrothermal system is directly linked to the thermal and tectonic activity of the Emeishan Large Igneous Province (ELIP). Deep-seated basement faults, reactivated during the ELIP event, acted as the primary conduits for hydrothermal fluid migration. This study also shed light on the regional characteristics and formation mechanisms of dolomites influenced by LIPs or abnormal tectonic activities.

**Data availability statement:** All relevant data are within the manuscript and its Supporting information files.

**Funding:** This research was funded by the National Natural Science Foundation of China (Grant No. U19B6003). This work was also supported by the research group "Geologia Sedimentària" (2021 SGR-Cat 00349) funded by the Generalitat de Catalunya, and the DGICYT Spanish project PID2021-122467NB-C22, the Natural Science Foundation of Chongqing Municipality (CSTB2023NSCQ-MSX1022), and the Natural Science Foundation of Sichuan Province (2022NSFSC1177).

**Competing interests:** The authors have declared that no competing interests exist.

## 1. Introduction

Structurally controlled dolomitization is a significant diagenetic process in many sedimentary basins, often linked to the circulation of hot, saline brines that can create economically important hydrocarbon and mineral resources [1–3]. The term "hydrothermal dolomite" (HTD) is widely used to describe these products. However, the term "hydrothermal" itself has been applied inconsistently. Following the time-honored and rigorous definition by White [4], and strongly advocated for by Machel and Lonnee [5], we define a hydrothermal process as one involving 'aqueous solutions that are warm or hot relative to its surrounding environment'. This distinction is critical, as the presence of high-temperature dolomite fabrics, such as coarse non-planar crystals and saddle dolomite cements, does not in itself prove a hydrothermal origin without knowledge of the host rock's thermal history. Hydrothermal dolomites are generally featured by medium to coarse curved crystals, saddle cement fillings, breccia and zebra textures, high temperatures and salinities, etc. [6–10]. Recent studies suggest that the thermal convection of seawater or deep fluids in high permeable strata could also explain the formation of the dolomites associated with abnormal geothermal activities, which are interpreted as thermal convection dolomitization [11–18]. Thermal convection dolomites are mainly featured by high-temperature seawater-derived diagenetic fluids and a scarcity of typical features of hydrothermal dolomites [2,15,19]. Here, the main types of dolomites in the Middle Permian in the Sichuan Basin (SCB) show broadly similar characteristics to those of the hydrothermal dolomites and thermal convection dolomites summarized above, suggesting a possible relationship with the activities of Emeishan Large Igneous Province (ELIP) event.

In recent years, the Middle Permian dolomites have been the critical targets for carbonate reservoirs exploration in the SCB due to the successful exploration of natural gas [20–23]. Numerous studies have been done on the dolomites, especially on the rock types, source of dolomitized fluids, and dolomitization mechanisms [24–30]. Various types of dolomites have been described, including planar-e (euhedral) to planar-s (subhedral) dolomites with fine to medium crystals (Rd1 and Rd2 following…..), non-planar dolomite with medium to coarse crystals (Rd3), and saddle dolomite cement (Sd) [14,31]. It is accepted that Rd1 and Rd2 were formed by coeval Permian seawater during the penecontemporaneous to shallow burial period [13,29,30]. Typically, the Rd2 in northwestern Sichuan Basin was formed by the thermal convection dolomitization at the platform margin shoals due to the convection exchange between warm pore seawater and cold deep seawater from outside the slope [13]. Furthermore, most studies believe that Rd3 and Sd were originated from high-temperature and high-salinity hydrothermal fluids, especially in the southwestern and central SCB [25,30,32]. However, most studies mainly focused on examining the properties of dolomitized fluids and recognizing the dolomitization mechanism locally. The relationship between dolomitized fluids and ELIP and the controlling factors of the main dolomite types of the whole basin are less known because the lack of comparison of the dolomitized fluid properties of Rd3 and Sd in different regions in the SCB basin (SW, central, and NW).

While previous studies have locally identified hydrothermal dolomites in the Sichuan Basin, a systematic, basin-scale comparison of their fluid properties and their

relationship to the ELIP has been lacking. Most interpretations have relied on localized data, making it difficult to build a cohesive, basin-wide genetic model. The key research gap lies in understanding how a large-scale thermal anomaly, such as a Large Igneous Province, manifests in the diagenetic record across a basin over hundreds of kilometers, especially in a basin now known to have experienced multiple, distinct hydrothermal pulses [33,34]. This study addresses this gap by integrating petrographic and geochemical data from three distinct regions of the basin—southwestern (proximal to the ELIP), central (intermediate), and northwestern (distal). Our goal is to test the hypothesis that there is a predictable geochemical gradient in dolomitizing fluids radiating away from the thermal center of the Permian-aged event. By doing so, we aim to: i) characterize the petrography of the key dolomite phases (Rd3 and Sd); ii) constrain their formation temperatures and fluid compositions using fluid inclusion thermometry and stable isotopes; iii) systematically compare these diagenetic fluid properties across the basin; and iv) develop a comprehensive genetic model that directly links the spatial heterogeneity of this specific hydrothermal event to the influence of the ELIP. The findings have broader implications for predicting reservoir quality in other basins affected by large-scale tectono-thermal events.

## 2. Geological background

The Sichuan Basin (SCB) is a typical craton basin located on the western margin of the Yangtze Block (Fig 1a). A lot geological events were recorded in the basin during the Permian period, including the activities of the ELIP [35–37], the expansion of the Paleo-Tethys Ocean [38–40], the Permian Chert Event (PCE) [41–43], and the Guadalupian mass extinction [44–47].

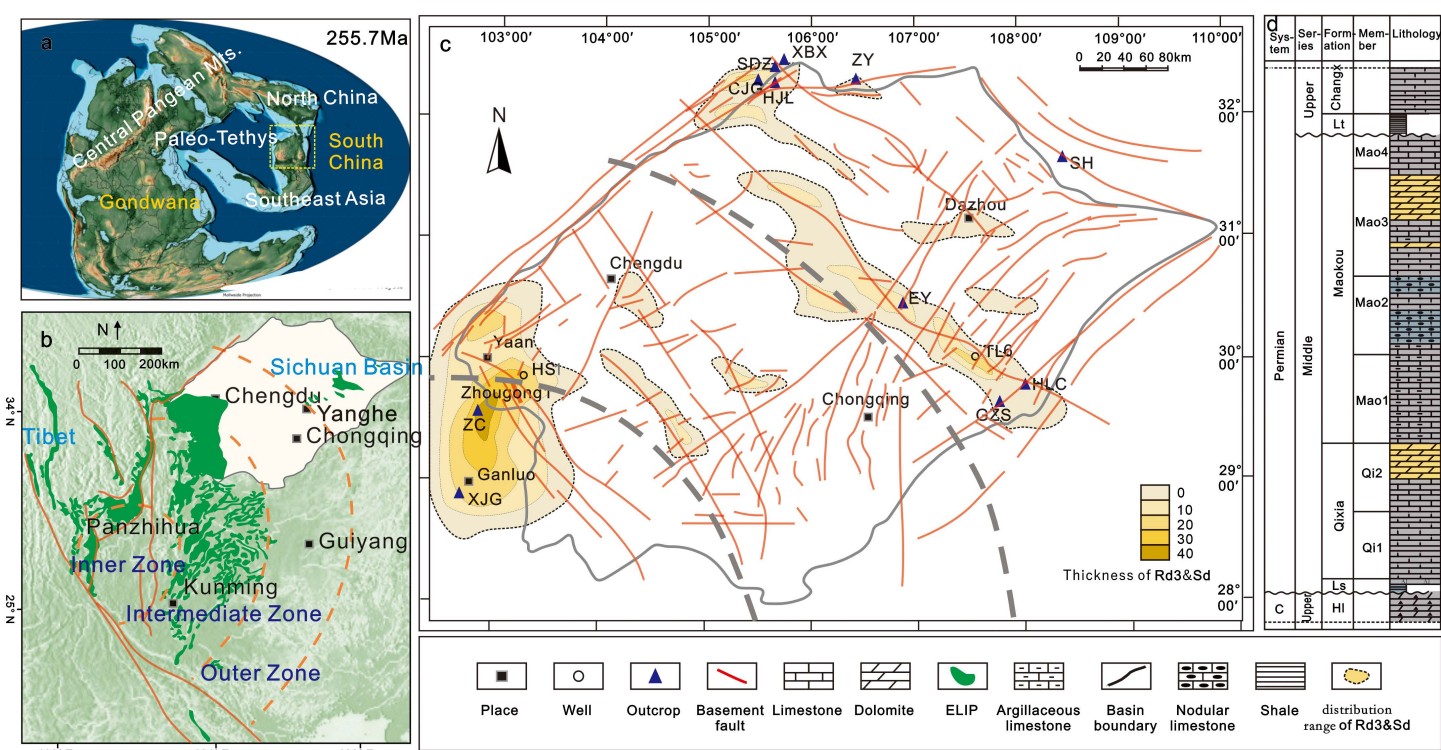

**Fig 1. Geological structural background of the study area.** (a) Global paleogeography and the location of SCB during Middle Permian (~255.7 Ma) [48]. (b) The distribution map of ELIP basalts and the location of the Sichuan Basin [37,49,50]. (c) The location of study sections, wells, the deep basement faults, and Rd3 and Sd distributions of the Middle Permian in the SCB [51,52]. (d) Stratigraphic column of the Middle Permian Formation in the SCB.

The activitiy of the ELIP occurred in the Middle-Late Permian, and the main phase of eruption took place for a short time (~1 Ma) [37,49]. ELIP is mainly located on the western edge of the Yangtze Platform, bounded by the Ailaoshan-Honghe fault zone and the Longmenshan rift zone (Fig 1b) [37,49,50]. The significant regional extensional tectonic movements were caused by the uplift of the Emeishan mantle plume and the subduction of the Mianlue Ocean, accompanied by the reactivation of the original basement faults (Fig 1c) [37,51–53]. However, only part of the deep faults was activated during this period [52,54]. Besides, abnormally high geothermal conditions persisted for tens of Ma and reached a maximum of 120 mW/m² of the paleo-heat flow at the ELIP eruption (~259Ma) [50,55,56].

The Middle Permian strata in the SCB include the Liangshan Fm., Qixia Fm., and Maokou Fm. from bottom to top (400–500 m) (Fig 1d). The Liangshan Fm. is composed of coastal sediments, including mudstones, siltstones, and sandstones. The Qixia Fm. includes two members. The lower member is composed of medium to thin layers of bioclastic limestones, siliceous limestones, bedded cherts, and nodular cherts. The upper member is mainly made of medium to thick layers of bioclastic limestones and dolomites with the characteristics of shallow water and high-energy depositional conditions. The Maokou Fm. comprises four members. Nodular limestones, consisting of interbedded bioclastic micrites and argillaceous limestones, are widely developed in the lower member, representing a deep-water sedimental environment. The second and third members are mainly composed of bioclastic limestones and dolomites, and nodular cherts are developed in some areas. The upper member is mainly made of bioclastic limestones, argillaceous limestones, and cherts, with internal erosive surfaces due to the influence of the ELIP tectonic movements.

## 3. Samples and methods

The dolomite and limestone samples were collected from drilling wells and field outcrops of different regions in the SCB (Fig 1c). Fieldwork did not involve endangered species, protected areas, or sensitive/private information, therefore no additional institutional permissions were necessary. Core observation and sampling were carried out under the written permission granted by the Exploration Branch of China Petroleum and Chemical Corporation (Chengdu, China). Based on field and drilling core observations, thin sections were elaborated and analysed with polarized light petrographic microscopy at the School of Earth and Space Sciences (Peking University). Thin sections were stained with Alizarin Red S to distinguish dolomites from calcites [57]. Detailed characteristics of dolomite samples were investigated in double polished thin sections and bulk samples (1 cm³) w by scanning electron microscopy (SEM) using an FEI-QUANTA650FEG. The elemental composition of various minerals was determined by spectroscopic testing.

Geochemical analyses were carried out at the Key Laboratory of Orogenic Belts and Crustal Evolution at Peking University. Carbon and oxygen isotopic values of 23 limestones and 47 dolomite samples were tested with the IsoPrime 100 instrument. A weight of 200 mg of each powder sample obtained by a micro-drill was tested after dissolution in 99% $H_3PO_4$. The standard sample in this test is IAEA CO-8 calcite with an analytical precision of ±0.1‰. Strontium isotope analyses of 11 limestone samples and 34 dolomite samples were tested with a TRITON mass spectrometer. The $^{87}Sr/^{86}Sr$ ratios were tested after 120 mg powder sample was dissolving in 2.5 mol/L HCl solution, and then the supernatant was centrifuged and passed through a cation exchange column, using HCl as an eluent to separate pure strontium. The results for Sr isotopes were corrected with the NBS987 standard, and the mean analytical precision was ± 1.0 × 10⁻⁵ (2δ).

The fluid inclusion (FIs) study was carried out at the University of Regina, SK, Canada. The homogenization temperature (Th) and final melting temperature (Tm-ice) of two-phase aqueous inclusions in thin sections (50–100 µm thick) were recorded using a Linkam THMGS600. The fluid inclusion measurement and data analysis are based on the principle of fluid inclusion assemblage (FIA) [58], which can eliminate the interference of secondary inclusions and increase the reliability of inclusion data. Tm-ice was measured by cyclic thermometry with an accuracy of ± 0.1°C. A total of 145 fluid inclusions (including 27 FIAs) were tested for microscopic thermometry in this study (See S1 Table, refer to Appendix S1.xlsx). The calculation of salinity was used by the calculation formula established by Steele-MacInnis et al [59,60]. The oxygen

isotopic compositions (δ¹⁸O fluid) were initially calculated using the mean fluid inclusion homogenization temperatures (Th) and the corresponding δ¹⁸O values of the dolomites, based on the equilibrium fractionation equation of Horita [61].

## 4. Results

### 4.1. Petrography

The Middle Permian dolomite in the SCB comprises three categories of replacement dolomites (Rd1, Rd2, and Rd3) and one saddle dolomite cement (Sd). Rd1 is mainly developed in the SW region, followed by the NW region, and is rarely found in the central SCB. The Rd1 is composed of fine (<50 μm) and planar to non-planar crystals with well-preserved original limestone fabric (Fig 2a). Rd2 is widely distributed in the basin with similar characteristics in different regions. The Rd2 is composed of fine to medium crystals (50μm∼250μm) with plane-e (euhedral) to plane-s (subhedral) crystals and

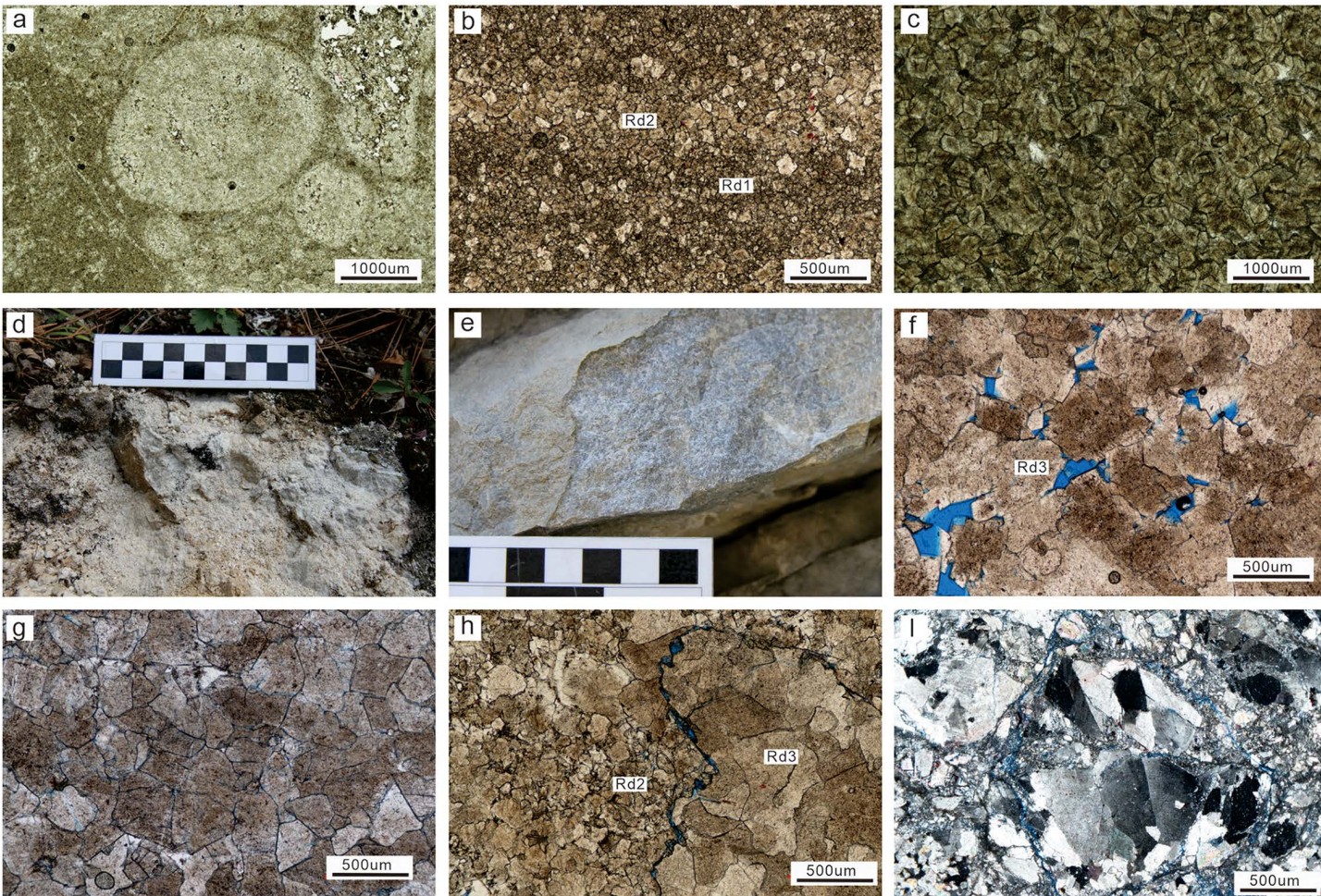

**Fig 2. Photographs showing the petrographic characteristics of replacement dolomite (Rd1 – Rd3).** (a) Very finely crystalline Rd1 with original residual fabric (XJG; $P_2m$). (b) Rd2 dolomite crystals were recrystallized from Rd1 dolomite (XJG; $P_2q$). (c) Euhedral to subhedral crystalline Rd2 have a foggy core surrounded by a clear rim (XJG; $P_2q$). (d) Field photo showing grey-dark grey sucrosic Rd3 dolomite with vugs (CJG; $P_2q$). (e) Field photo showing grey sucrosic Rd3 dolomite (XJG; $P_2q$). (f) Rd3 with medium to coarse non-planar crystals (XJG; $P_2q$). (g) Coarse crystalline Rd3 showing mosaic contact (EY; $P_2m$). (h) Fine crystalline Rd2 recrystallized to Rd3 with coarse and curved crystals (XJG; $P_2m$). (i) Coarse crystalline Rd3 display undulous extinction (SDZ; $P_2q$).

dark-core and bright-rim textures (Fig 2c). The original fabric of Rd2 is poorly preserved although some mimetic fabric of the bioclasts are still preserved.

Rd3 is primarily distributed in the SW, NW, and central Sichuan Basin and displays relatively similar petrological characteristics. The hand specimens of Rd3 are grey-dark to grey sucrosic dolomite (Fig 2d and 2e). The crystals of Rd3 are larger than those of Rd2, with medium to coarse size (250 μm~2 mm) and non-planar shape (Fig 2f and 2h). There is no obvious cloudy-core and clear-rim texture in Rd3, and undulatory extinction can be seen broadly under cross-polarized light (Fig 2i). The original fabric in Rd3 is completely destroyed, and its crystals are mosaic contact due to the overgrowth of dolomite crystals (Fig 2f and 2h). Sd is widely distributed in the Middle Permian strata in the SCB were it fills fractures and vuggy porosity (Fig 3a–3c). Sd often is milky white and coexists with dark surrounding rocks to present a "zebra texture" structure (Fig 3a and 3c), especially in the SW and central SCB. The crystal size of Sd are larger than that of Rds, and the crystal shapes are non-planar. Besides, Sd exhibit strong undulous extinction (Fig 3f).

## 4.2. Microthermometry and salinity

Most of the FIs observed in this study appeared as isolated and FIAs (Fig 4). For Rd3, the Th values are in the range of 98–248°C (average 136.2°C). The Tm-ice of Rd3 is from –25.3 to –3.1°C (average –9.6°C), and the calculated salinities range from 5.1 to 24.8wt% (12.3 wt% on average) (S1 Table). The Th values of Sd range from 96°C to 255°C (average 156.2°C), the Tm-ice values are from –25.3 to –2.2°C (average –9.4°C), and the salinities range from 3.7 to 24.8wt % (average 12.6wt%) (S1 Table).

The temperatures and salinities in different areas of the SCB vary. In the SW SCB, Th values of Rd3 range from 182 to 248°C (average 207.4°C), and salinities range from 9 to 24.8wt% (average 17.3wt%). The Th values of Sd range from 54 to 255°C (average 204°C), and the salinities range from 4.1 to 24.8wt% (average 14.3wt%). In the central SCB, Th values of Rd3 are much lower than those in the SW region, with the range of 107.4 to 147.4°C (average 131°C), and the salinities are slightly lower than those in the SW region with a range of 5.71 to 21.89wt% (average 15.2wt%). Similarly, the Th values and the salinities of Sd range from 114.5 to 187°C (average 137°C) and 3.71 to 22.04 wt% (average 13.0 wt%). In the NW SCB, the Th values of Rd3 are the lowest in the entire basin with the range of 98–130°C (average 114.2°C), the

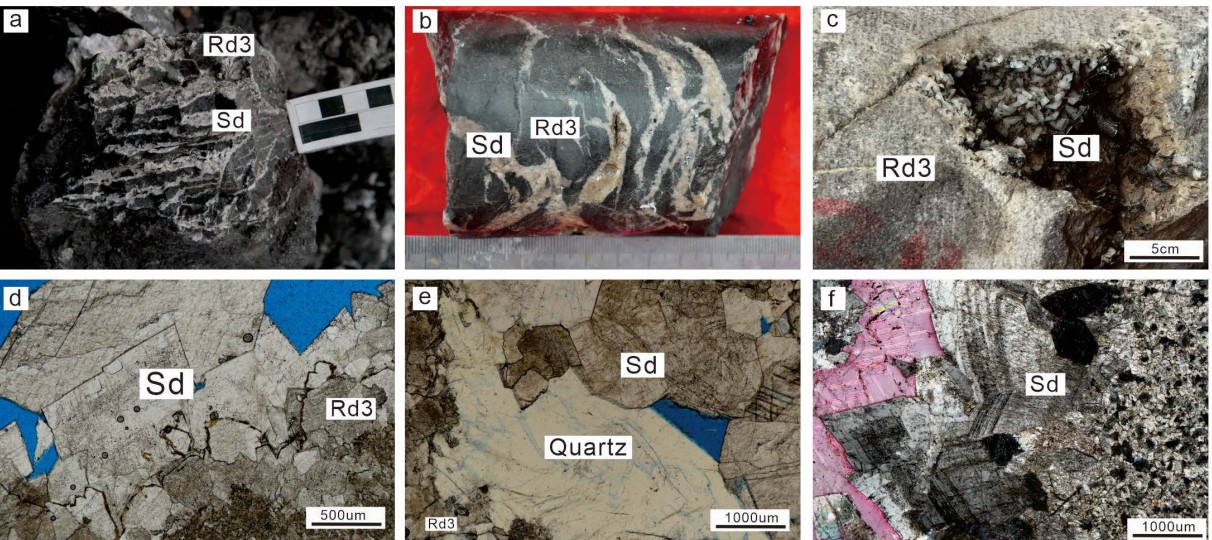

**Fig 3. Photographs showing the petrographic characteristics of saddle dolomite cement (Sd).** (a) Zebra pattern composed of Rd3 and Sd (ZC; P$_2$m). (b) Milky white Sd fill in the fractures (TL6; P$_2$m). (c) Milky white Sd fill in the dissolved pores (MX42; P$_2$q). (d) Rd3 and Sd are cut by stylolites (XJG; P$_2$m). (e) Quartz filled in the pores of Rd and Sd (XJG; P$_2$q). (f) Sd exhibits strong undulatory extinction under cross-polarized light (CJG; P$_2$q).

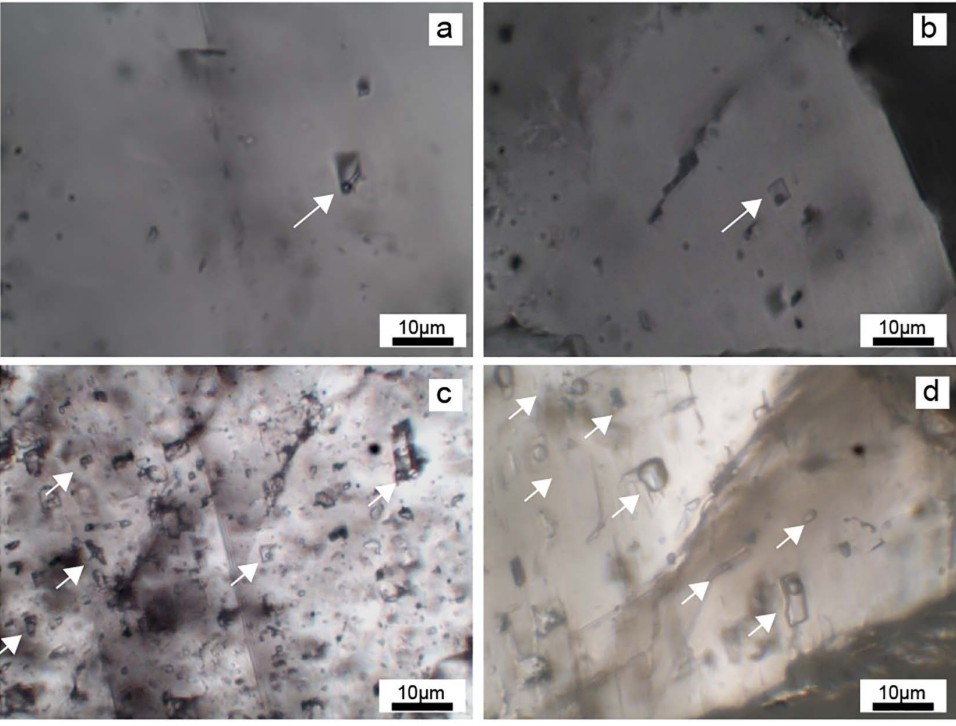

**Fig 4. Photomicrographs showing fluid inclusions petrography. a-b.** Irregular distribution of isolated FIs in dolomite crystals (XJG; $P_2q$); c. FIs outline the growth zone of Sd dolomite (TL6; $P_2m$); d. FIs outline the growth zone of Sd dolomite (XJG; $P_2q$).

salinities are also the lowest with the range of 5.1 to 9.7 wt% (average 7.6 wt%). For Sd, the Th values and salinities are from 96 to 135°C (average 116.8°C) and 6.6 to 10.4 wt% (average 8.5 wt%). To sum up, the fluid inclusions in the Rd3 and Sd samples in the SW SCB have the highest Th values and salinities among the three regions, followed by the central SCB and the NW SCB (S1 Table; Fig 5a; Fig 5b).

## 4.3. Carbon and oxygen isotopes

Overall, the $\delta^{18}O$ and $\delta^{13}C$ values of Rd3 range from −12.01 to −6.5‰ (average −9.17‰) and from 0.6 to 5.28‰ (average 3.31‰), respectively. The $\delta^{18}O$ and $\delta^{13}C$ values of Sd are in ranges of −12.31 to −6.9‰ (average −9.07‰) and 2.09 to 3.86‰

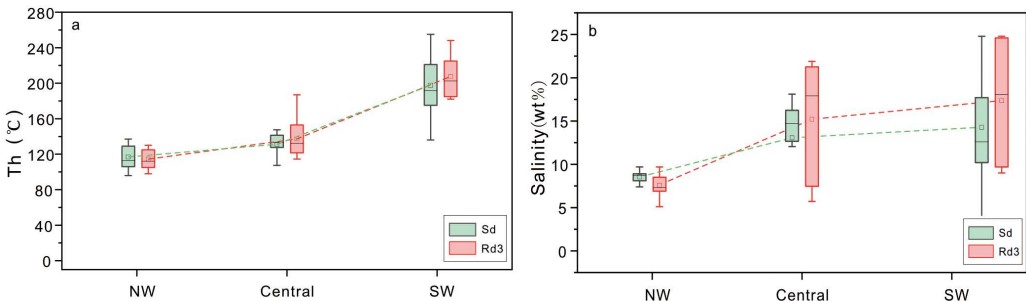

**Fig 5. Homogenization temperature and salinity of dolomite fluid inclusions. (a) The Th profile of Rd3 and Sd from three different areas in the Sichuan Basin. (b)** The salinity profile of Rd3 and Sd from three different areas in the Sichuan Basin.

(average 3.18‰). The $\delta^{18}O$ and $\delta^{13}C$ values of limestone samples are from −7.8 to −3.7‰ (average −5.6‰) and from 2.6 to 5.15‰ (average 3.6‰). Compared with the relatively similar $\delta^{13}C$ values, the $\delta^{18}O$ values of Rd3 and Sd in different regions are different (See S2 Table, refer to Appendix S2.XLSX; Fig 5). The $\delta^{18}O$ values of Rd3 and Sd in SW SCB are from −12.0 to −9.1‰ (average −11.4‰) and from −12.3 to −10.7‰ (average −11.6‰), which are the lowest in the whole basin. In the central SCB, the $\delta^{18}O$ values of Rd3 and Sd range from −8.4 to −6.8‰ (average −7.9‰) and −8.2 to −7.0‰ (average −7.8‰), which are slightly higher than those in SW SCB. The $\delta^{18}O$ values of Rd3 and Sd samples in the NW SCB span from −7.4 to −6.5‰ (average −6.9‰) and from −7.4 to −6.9‰ (average −7.1‰), which are the highest of the SCB (S2 Table; Fig 6).

The $\delta^{18}O$ values of the dolomitization fluids range from 5.47 to 6.8‰ (average 6.31‰) in NW SCB, 6.36 to 8.0 ‰ (average 7.02‰) in central SCB and 6.94 to 9.66 ‰ (average 8.56‰) in SW SCB (S2 Table; Fig 7).

### 4.4. Radiogenic Sr isotopes

The $^{87}Sr/^{86}Sr$ ratios of Rd3 and Sd range from 0.70738 to 0.70941 (average 0.70837) and 0.70764 to 0.71018 (average 0.70859). The $^{87}Sr/^{86}Sr$ ratios of limestone samples are from 0.7053 to 0.7089 (average 0.70747) (S2 Table; Fig 8). The $^{87}Sr/^{86}Sr$ ratios of Rd3 and Sd in SW Sichuan Basin range from 0.70823 to 0.70941 (average 0.70870) and 0.70837 to 0.71018 (average 0.70924), which are higher than those of Permian seawater (0.70680–0.70807) [62,63]. In the central SCB, the $^{87}Sr/^{86}Sr$ ratios of Rd3 and Sd range from 0.70801 to 0.7091 (average 0.70846) and 0.7083 to 0.7092 (average 0.70861), which are also higher than those of Permian seawater but slightly lower than those of SW Sichuan Basin. The $^{87}Sr/^{86}Sr$ ratios of Rd3 and Sd in the NW Sichuan Basin are from 0.70738 to 0.7084 (average 0.70784) and 0.70764 to 0.70779 (average 0.70771), which are basically in the range of $^{87}Sr/^{86}Sr$ ratios of Permian seawater (Fig 8).

## 5. Discussion

### 5.1. Diagenetic sequence and distribution characteristics

Based on cross-cutting relationships, the paragenetic sequence between the four dolomite types (Rd1, Rd2, Rd3, and Sd) and related cement (including quartz cement, calcite cement) has been stablish (Fig 9). Recrystallization of Rd2 from Rd1 (Fig 2b) supports that Rd2 was formed later than Rd1. Similarly, Rd2 with planar-e to planar-s fine crystals evolved to Rd3

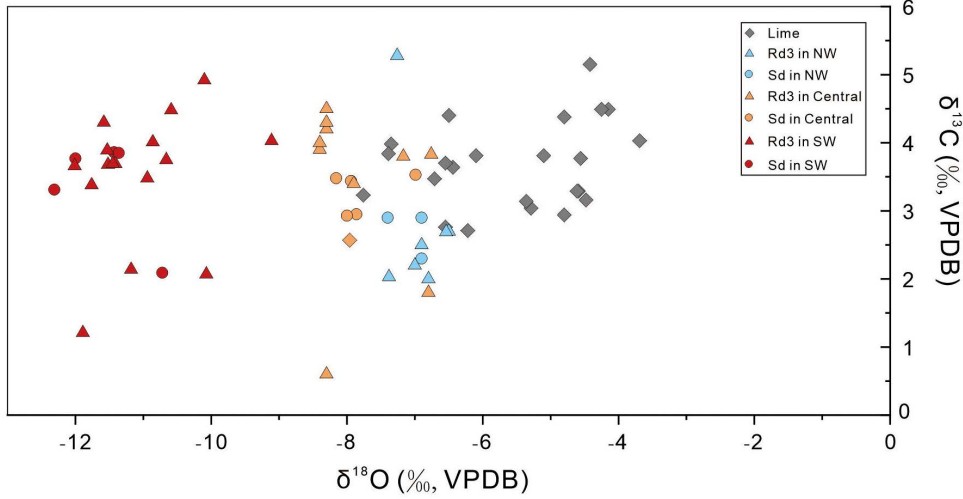

**Fig 6. $\delta^{13}C$ versus $\delta^{18}O$ values of different dolomites and hist rocks in the three regions.**

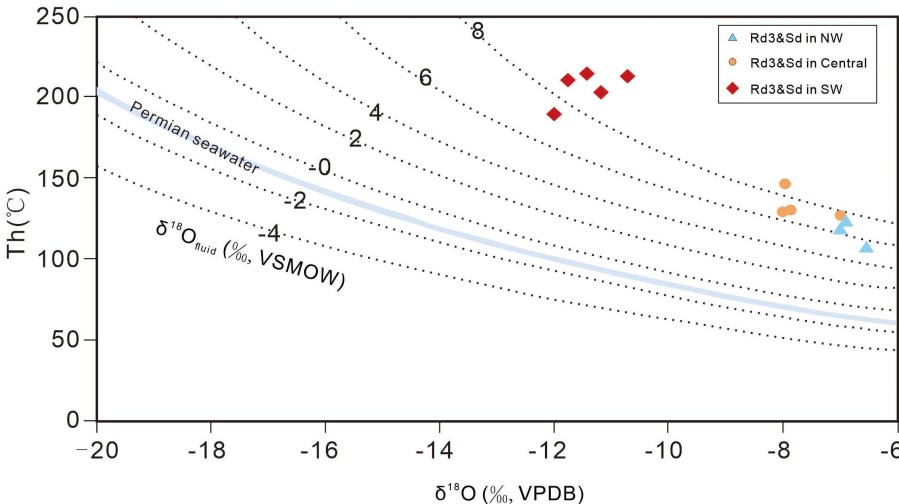

**Fig 7. δ13C versus δ18O values of the dolomitization fluids in the three regions.**

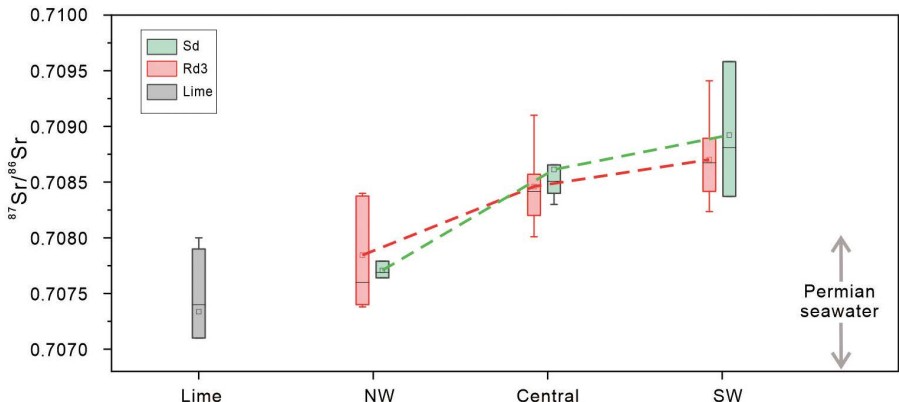

**Fig 8.** 87Sr/86Sr **ratios of limestones, Rd3, and Sd in the three regions.** 87Sr/86Sr ratios of Permian seawater range from 0.70680 to 0.70807 [62,63].

with non-planar medium to coarse crystals (Fig 2h), indicating that Rd3 formed after Rd2. The non-planar and sweeping extinction features of Rd3 crystals (Fig 2i) suggest that it has experienced intereaction with high temperature, exceeding the "critical roughening temperature" (CRT), which is the temperature at which a crystal surface undergoes a phase transition from a smooth (faceted) surface to a rough surface, or high saturated dolomitized fluids [64–66]. Sd is found in the pores and fractures of replacive dolomites (Rd1, Rd2, and Rd3) (Fig 3) and therefore the formation of Sd was later than that of three replacive dolomite types. Furthermore, Rd3 and Sd are commonly associated, suggesting that the formation process of these two kinds of non-planar dolomites was continuous [67,68]. Formation of the four types of dolomites occurred in a shallow burial environment (<500m) because stylolites truncated through the crystals of Rd3 and Sd (Fig 3d) [69]. In addition, quartz cement fills the intercrystalline pores of Rd3, indicating the later formation time of quartz (Fig 3e). Most studies imply that the siliceous materials in the Middle Permian are of hydrothermal origin [41–43].

Despite Rd3 and Sd owning similar petrological characteristics in different regions, the temporal and spatial distributions are heterogeneous among the SW, central, and NW SCB. In the SW SCB, a large amount of Rd3 is distributed in

| DIAGENETIC EVENTS | EARLY ———————▶ LATE |
|---|---|
| Micritic replacive dolomite (Rd1) | ▭ |
| Fine-grained replacive dolomite (Rd2) | ▭ |
| Medium to coarse replacive dolomite (Rd3) | ▭ |
| Hydrofracturing fractures | ▭ |
| Saddle dolomite cement (Sd) | ▭ |
| Quartz cements (Qz) | ▭ |
| Calcite cements in fractures (Cc) | ▭ |
| Burial stylolite | ▭ |

**Fig 9. Diagenetic sequence of the dolomites in SCB.**

both Qixia and Maokou Fm. with an abundance of more than 50% of the whole strata. Moreover, Sd is very common in fractures, pores, and vugs. The developments of Rd3 and Sd are always associated with deep faults. For example, in the Well ZG1, Well HS1, and ZC Outcrop proximal to the basement fault, the thickness of dolomites is larger (greater than 20 m) in the areas proximal to the deep faults (e.g., Well ZG1, Well HS1, and ZC section) than those of the areas away from the deep faults (e.g., XJG section). In the central Sichuan region, Rd3 and Sd are mainly concentrated near the basement faults in the upper part of the Mao2 and Mao3 members with thickness less than 20 m. In the NW SCB, the abundance of Rd3 and Sd are much lower than those of the SW and central SCB, and they generally occur adjacent to Rd2, which is the primary dolomite type in the northwestern region (Fig 1C) [13,14]. In general, the abundance of Rd3 and Sd was high in SW SCB, medium in central SCB, and low in NW SCB.

### 5.2. The dolomitized fluids of Rd3 and Sd

It is generally believed that Rd1 was formed through the evaporative seawater during the penecontemporaneous stage, and Rd2 was mainly formed through seawater driven by thermal convection during the shallow burial stage [8,70,71]. The formation of Rd3 and Sd is overwhelmingly recognized as hydrothermal dolomitization [72–74], except for the thermal convection dolomitization in the NW SCB [13,14]. However, there are still many differences in the properties of Rd3 and Sd dolomitized fluids among different regions.

Multiple pieces of evidence indicate that Rd3 and Sd are formed under high temperature conditions although the formation temperature varies along the basin. Compared with the planar-e to planar-s crystals of Rd1 and Rd2, the crystals of Rd3 and Sd dolomite are curved, suggesting that the crystal growth rate was faster and the temperature of the diagenetic fluids was higher [64,65]. The rapid crystal growth is also supported by the abundant FIs observed in the Rd3 and Sd crystals [66,75]. The $\delta^{18}O$ values of Rd3 and Sd samples (average $\delta^{18}O_{Rd3} = -9.17\%$, average $\delta^{18}O_{Sd} = -9.07\%$) are lower than those of Rd1 (average –6.68‰) and Rd2 (average –5.82‰) [30], also suggesting that Rd3 and Sd formed at higher temperatures [76–79]. Furthermore, the high Th of Rd3 and Sd (average Th-Rd3 = 136.2°C, average Th-Sd = 156.2°C) prove the high temperature of dolomitizing fluids of Rd3 and Sd, which is much higher than the regional maximum temperature under the burial conditions (110°C) [30]. Besides, the temperature of dolomitizing fluids of Rd3 and Sd was higher in SW and central Sichuan Bain than those in NW Sichuan Basin (S1 Table; Fig 5).

The salinities of Rd3 and Sd also varies significantly between the different regions. The salinities of FIs in most of the Rd3 and Sd crystals average 13.9wt%, much higher than those of Permian seawater [80,81]. However, no evaporite minerals were found in the SCB during this period, and the $\delta^{18}O$ values of carbonate minerals are generally negative, excluding the contribution of contemporaneous evaporation to the high salinities. Therefore, we suggest that the high salinities were caused by fluids with hydrothermal properties, combined with the high-temperature features. However, most salinities of the

FIs in the Rd3 and Sd samples from NW SCB are similar to those of Permian seawater, much lower than those of SW and central SCB (Fig 5b), suggesting that the dolomite fluids in this area suffered few influences from hydrothermal fluids.

$^{87}Sr/^{86}Sr$ ratios and special mineral cement of Rd3 and Sd can further indicate the source of dolomitized fluids because Sr isotopic composition is mainly controlled by the source of strontium due to the stability of $^{87}Sr/^{86}Sr$ ratios during carbonate diagenesis [44,82–85]. Generally, the 87Sr/86Sr ratios of Rd3 (average 0.70870) and Sd (average 0.70924) are much higher than those of Permian seawater (0.70680–0.70807) and host limestones (average 0.7073) [62,63]. The high value of radiogenic Sr is likely associated with the circulation of hydrothermal fluids through the underlying clastic strata [3,86,87]. In addition, a large number of hydrothermal minerals (fluorite, quartz, siderite) are found in these dolomites (Fig 10a–c), further supporting the evidence of hydrothermal fluids. Overall, the diagenetic fluids of Rd3 and Sd in southwestern and central SCB are characterized by high temperature, high salinity, high $^{87}Sr/^{86}Sr$ ratios, and enrichment of hydrothermal minerals indicating the significant contribution of hydrothermal fluids for the origin of Rd3 and Sd. However, the dolomitized fluids of Rd3 and Sd in NW Sichuan Basin are dominated by coeval seawater because the aforementioned low salinity samples from the NW region have similar $^{87}Sr/^{86}Sr$ ratios with those of Permian seawater and Permian limestones (Fig 8), and no hydrothermal minerals have been found.

Based on the corrected calculations, the oxygen isotopic compositions of the dolomitizing fluids ($\delta^{18}O_{fluid}$) provide a robust tracer for deciphering their origin and evolution (Fig 7). The values exhibit a clear and systematic spatial gradient, increasing from the northwestern (NW), through the central, and to the southwestern (SW) region of the Sichuan Basin (SCB). In the proximal SW region, closest to the ELIP's thermal center, the $\delta^{18}O_{fluid}$ values are the highest, ranging from +6.94‰ to +9.66‰ (VSMOW). These highly positive values are the classic signature of deep basinal brines that have undergone extensive, high-temperature isotopic exchange with $^{18}O$-rich silicate rocks of the crystalline basement or deep sedimentary strata. This provides compelling evidence for a fluid-buffered system dominated by the advective upwelling

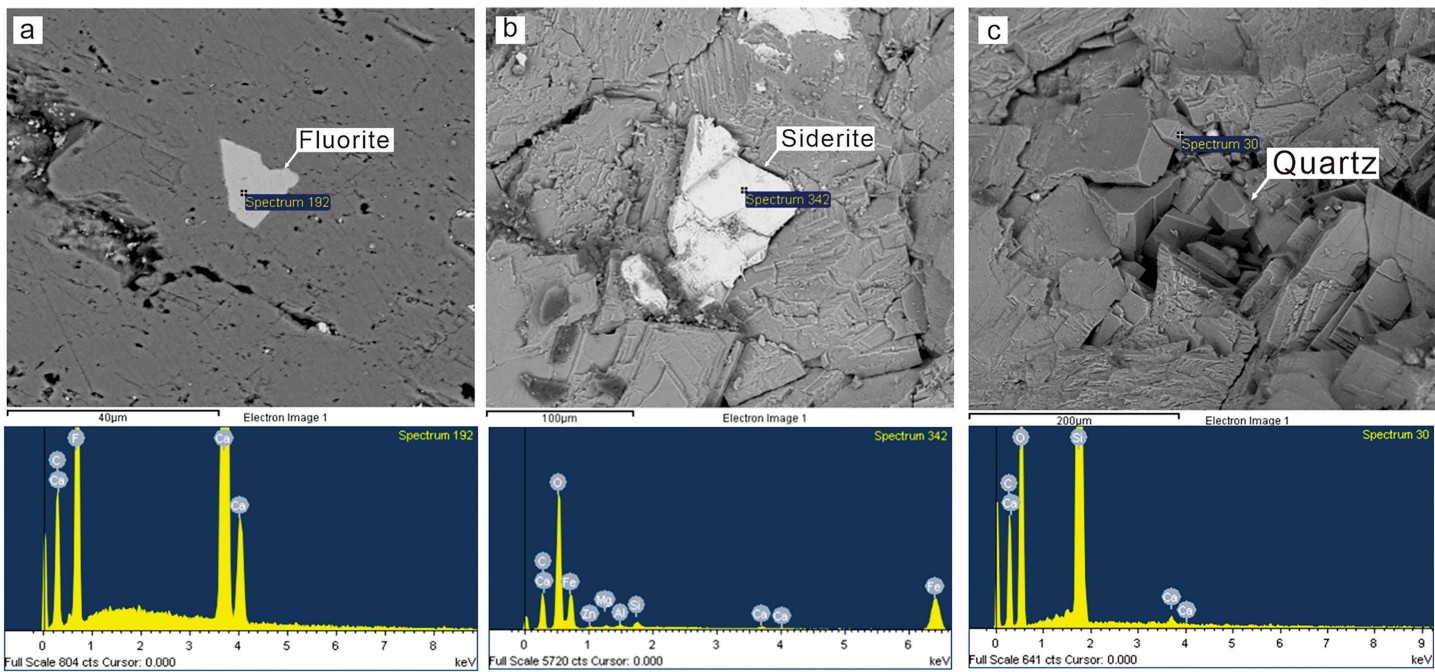

**Fig 10. Photographs of SEMS showing the hydrothermal minerals in Sd.** (a) Fluorite filled in Rd3 under SEM observation (EY; $P_2m$). (b) Siderite filled in Rd3 under SEM observation (TL6; $P_2m$). (c) Quartz filled in Rd3 under SEM observation (XJG; $P_2q$).

of deep, crustal-derived hydrothermal fluids along major fault conduits. In the distal NW region, the $\delta^{18}O_{fluid}$ values, while being the lowest in the study area, are still significantly positive, ranging from +5.47‰ to +6.80‰ (VSMOW). This signature suggests that the distal area was influenced by hydrothermal fluids that had either migrated a long lateral distance from the SW. The central SCB, with intermediate $\delta^{18}O_{fluid}$ values of +6.36‰ to +8.00‰ (VSMOW), represents a clear transitional and mixing zone. This systematic isotopic gradient across the basin powerfully traces the fluid pathway from a potent, deep-sourced hydrothermal system in the SW to a less evolved basinal fluid system in the NW, unequivocally linking the intensity and nature of the Permian hydrothermal event to the proximity of the ELIP.

In conclusion, the dolomitized fluids of Rd3 and Sd in the SW and central SCB are dominated by hydrothermal fluids as indicated by the high Th values, high salinity, high $\delta^{18}O$ fluid and high $^{87}Sr/^{86}Sr$ ratios. The dolomitized fluids of Rd3 and Sd in NW Sichuan Basin are most likely coeval Permian seawater, which is characterized by relatively low temperature, low salinity, low $\delta^{18}O$ fluid and low $^{87}Sr/^{86}Sr$ ratios.

## 5.3. Dolomitization mechanisms and implications for the ELIP control

The pronounced geochemical gradients observed across the Sichuan Basin—from lower temperatures, salinities, and near-seawater isotopic signatures in the northwest (NW) to significantly higher values in the southwest (SW)—suggest a continuum of fluid processes rather than two mutually exclusive dolomitization models. We propose an integrated model involving the interaction of two primary end-member fluids: (1) heated Permian seawater and (2) a deep, high-temperature, high-salinity basinal brine injected into the basin's SW margin via basement faults activated by the Emeishan Large Igneous Province (ELIP).

This spatial variation can be effectively understood through the concepts of rock-buffered versus fluid-buffered diagenetic systems, as recently proposed by Jiang et al. [88] for similar settings. In the distal NW region, far from the ELIP's magmatic center and major fault conduits, the system was largely rock-buffered. Here, lower fluid/rock ratios meant that the chemistry of the dolomitizing fluids—likely seawater heated via thermal convection due to the elevated regional geothermal gradient—was strongly controlled by interactions with the host limestone. This resulted in dolomites (Rd3 and Sd) that retained geochemical signatures closer to the host rock, including less radiogenic $^{87}Sr/^{86}Sr$ ratios and lower overall temperatures and salinities.

In contrast, the proximal SW region was intensely fluid-buffered. The system here was dominated by the high-flux, advective influx of hydrothermal brines ascending directly along major basement faults. The high fluid/rock ratio ensured that the geochemical signature of the external fluid overwhelmed that of the precursor limestone. This process imparted a strong crustal signature on the resulting dolomites, evidenced by their very high temperatures, high salinities, highly radiogenic $^{87}Sr/^{86}Sr$ ratios from interaction with underlying siliciclastic or basement rocks, and enriched $\delta^{18}O$ fluid values. The central SCB represents a transitional mixing zone between these two end-members, exhibiting intermediate geochemical properties. This integrated model, which frames thermal convection as the distal, rock-buffered expression of a large, fault-focused, fluid-buffered hydrothermal system, elegantly unifies the diverse observations across the basin and provides a robust mechanism for the observed geochemical gradients.

The formation and heterogeneous distribution of the replacive dolomite (Rd3) and saddle dolomite (Sd) across the Sichuan Basin require a large-scale, high-flux fluid and thermal driver, pointing directly to major tectono-thermal events. While both the Emeishan Large Igneous Province (ELIP) event and the later Longmenshan Orogeny have been proposed as potential drivers, recent advances in in-situ carbonate U-Pb geochronology provide a powerful tool to distinguish their respective impacts in time and space. The geochronological data from across the basin, when integrated with our geochemical gradients, overwhelmingly supports a direct genetic link between the Permian-aged hydrothermal dolomitization event and the ELIP.

Crucially, U-Pb dating from different parts of the basin reveals distinct temporal domains of hydrothermal activity. In the southwestern Sichuan Basin, proximal to the ELIP's eruption center, recent studies have firmly constrained the timing of hydrothermal dolomite formation to the Middle-Late Permian. Zou et al. [34] dated saddle dolomites in the Qixia Formation

from the PR1 well (located in the ELIP intermediate zone) to 257.9–251.0 Ma, which directly overlaps with the main and late phases of ELIP magmatism (ca. 260–251 Ma). Similarly, Yang et al. [89] dated multiple dolomite and calcite cement phases in the southeastern Sichuan Basin to a range of 264 ± 10 Ma to 251 ± 11 Ma, again confirming a temporal link to the ELIP. These ages are consistent with the high temperatures (avg. Th > 200°C) and highly radiogenic $^{87}Sr/^{86}Sr$ ratios we observe in the SW region, reflecting an intense, fluid-buffered system driven by the ELIP's thermal and magmatic engine.

In stark contrast, U-Pb dating of similar hydrothermal dolomites from the northwestern Sichuan Basin, distal to the ELIP, yields significantly younger ages. Pan et al. [90,91] dated replacive and cement phases in the Maokou and Qixia formations of the NW Sichuan Basin to the Late Triassic (ca. 240–213 Ma). These authors explicitly link this later dolomitization event to fluid flow driven by the compressional tectonics of the Longmenshan Orogeny, an entirely separate geodynamic event. This temporal decoupling is critical: the NW region was not significantly impacted by the Permian hydrothermal pulse from the ELIP but was instead dolomitized by a later, unrelated system.

Therefore, the spatial distribution pattern of Rd3 and Sd described in our study—abundant in the SW, moderate in the Central, and sparse in the NW—is a direct reflection of the waning influence of the ELIP hydrothermal system with increasing distance from its center. The high abundance of Rd3 and Sd in the SW and Central regions is a direct consequence of intense, fault-focused fluid flow during the Permian event. The scarcity of these phases in the NW demonstrates that this region was beyond the primary reach of the ELIP's advective fluid system, where dolomitization was dominated by lower-temperature, rock-buffered processes (heated seawater convection) during the Permian, with more significant hydrothermal alteration occurring much later during the Triassic. This integration of geochronology and geochemistry provides a robust model that directly links the formation and distribution of the main Permian dolomite bodies in the Sichuan Basin to the ELIP activity.

## 5.4. Comparison with Global HTD Systems

The basin-scale geochemical gradients observed in the Sichuan Basin share striking similarities with other major HTD provinces, suggesting a common underlying mechanism for fluid flow and diagenesis. In the Western Canada Sedimentary Basin (WCSB), McCormick et al. [92] documented a clear evolution of zebra textures and dolomite geochemistry from the platform interior (distal) to the platform margin (proximal) relative to the fluid source along the Kicking Horse Rim. Distal dolomites are more stratabound and retain more primary features, while proximal dolomites are non-stratabound, pervasively recrystallized, and record higher fluid temperatures and more crustal-influenced geochemical signatures. This proximal-distal relationship is directly analogous to our findings, where the SW SCB represents the proximal setting (non-stratabound, intense hydrothermal alteration, fluid-buffered) and the NW SCB represents the distal setting (stratabound, lower temperatures, seawater-dominated, rock-buffered).

This pattern is not limited to these two basins. Furthermore, the fault-controlled hydrothermal systems in the Albian carbonates of the Basque-Cantabrian Basin, northern Spain, exhibit a similar zonation. Intense brecciation, boxwork textures, and high-temperature saddle dolomite are concentrated along major fault corridors, such as the Pozalagua fault, while dolomitization becomes progressively more stratabound and less intense away from these primary conduits [93,94]. Similarly, widespread hydrothermal dolomites in SW Sardinia, Italy, linked to late-Variscan fluid flow, show a comparable architecture. The most pervasive dolomitization and associated Mississippi Valley-Type mineralization are spatially restricted to the cores of the hydrothermal systems along major fault zones, grading into less altered facies distally [95] Furthermore, extensive studies in the Maestrat Basin, Spain, have similarly shown that syn-rift extensional faults acted as the primary conduits for hot, saline basinal brines, with the most intense alteration localized within fault zones, while pre-existing facies architecture and sequence stratigraphy controlled the lateral fluid flow and the final geometry of the dolostone bodies [96–98].

These examples from different continents and tectonic settings (extensional in the WCSB, strike-slip in Spain, post-orogenic in Italy) suggest that the model of a focused upwelling of hydrothermal fluids that mix with and displace ambient basinal waters along a lateral flow path is a common and fundamental mechanism in the formation of large-scale HTD bodies. Our study,

therefore, reinforces a universal model where the style and intensity of hydrothermal dolomitization is a predictable function of the distance from major, basement-tapping fault systems that act as the primary conduits for deep basinal brines (Fig 11).

## 6. Conclusion

This study provides a comprehensive, basin-scale model for the formation of Middle Permian hydrothermal dolomites (Rd3 and Sd) in the Sichuan Basin. Our key findings are as follows:

1) Multi-stage Diagenesis and Spatial Distribution: The main hydrothermal dolomites (Rd3 and Sd) post-date earlier, shallow-burial replacive phases (Rd1 and Rd2). Their distribution is highly heterogeneous, with the greatest abundance and most intense alteration occurring in the southwestern (SW) SCB, proximal to the Emeishan Large Igneous Province (ELIP) eruption center and major basement faults, and progressively decreasing through the central and northwestern (NW) regions.

2) A Fluid Continuum Explained by a Rock- vs. Fluid-Buffered Model: The observed geochemical gradients—including increasing temperature, salinity, $^{87}Sr/^{86}Sr$ ratios, and $\delta^{18}O_{fluid}$ values from the NW to the SW—are best explained by a fluid continuum model. This spatial variation reflects a transition from a distal, rock-buffered system in the NW, where dolomitization was driven by heated seawater via thermal convection, to a proximal, fluid-buffered system in the SW. The SW system was dominated by the advective influx of deep, crustal-derived hydrothermal brines that ascended along basement faults, imparting a strong crustal signature on the dolomites.

3) Direct Temporal Link to the ELIP: Recent U-Pb geochronology from multiple studies robustly links this specific hydrothermal event to the ELIP. The ages of hydrothermal dolomites in the SW and central SCB (ca. 264–251 Ma) directly overlap with the main phase of ELIP activity. This contrasts with younger, Late Triassic dolomitization in the NW SCB, which was driven by the separate Longmenshan Orogeny. Therefore, the distribution pattern of the Permian-aged dolomites investigated here is unequivocally controlled by the ELIP.

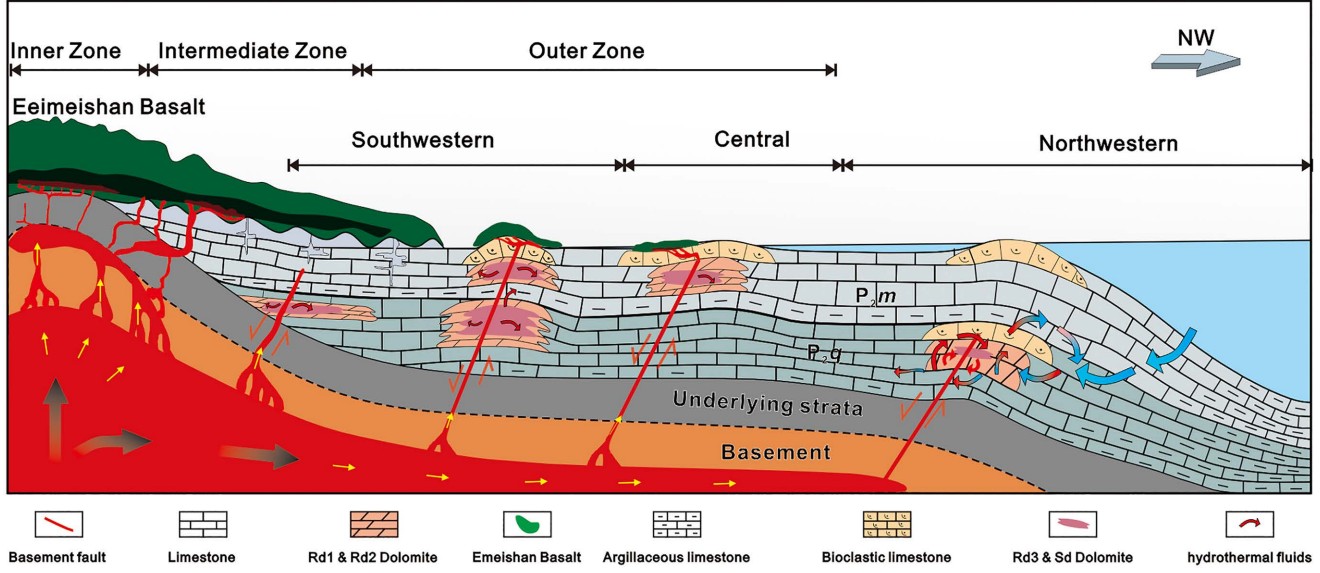

**Fig 11. A cartoon showing the mode of the dolomitization mechanism in this study is related to ELIP.**

4) Broader Implications for HTD Systems: The proximal-distal model developed for the Sichuan Basin, transitioning from fluid-buffered to rock-buffered domains, is analogous to other major HTD provinces worldwide, such as the Western Canada Sedimentary Basin and the Basque-Cantabrian Basin. This study reinforces a universal framework where the style, intensity, and geochemistry of hydrothermal dolomitization are predictable functions of the distance from major, basement-tapping fault systems that act as primary conduits for deep basinal brines during large-scale tectono-thermal events.

## Supporting information

**S1 Table. Original data table of fluid inclusions.**
(XLSX)

**S2 Table. Original data tables for carbon and oxygen isotopes and strontium isotopes.**
(XLSX)

## Acknowledgments

Guoxiang Chi and Hairuo Qing are thanked for their contribution to fluid inclusion analysis and the determination of the properties of dolomitized fluids. We also wish to thank the two reviewers for their valuable comments and suggestions.

## Author contributions

**Conceptualization:** Haofu Zheng, Xiong Duan.

**Data curation:** Jiajun He.

**Funding acquisition:** Xiong Duan, Bo Liu.

**Project administration:** Haofu Zheng, Xiong Duan.

**Writing – original draft:** Haofu Zheng.

**Writing – review & editing:** Haofu Zheng, Xiong Duan, Bo Liu, Anna Travé Herrero, Juan Diego Martín-Martín, Enrique Gómez Rivas.

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
