## [Decision Letter · Decision Letter 0]

30 Jul 2025

Dear Dr. Duan,

Thank you for submitting your manuscript entitled "Origins of the hydrothermal dolomites in Middle Permian, Sichuan Basin (SW China): Implication for the relationship with the Emeishan Large Igneous Province" to **PLOS ONE** .

Following a careful assessment by independent reviewers and thorough evaluation of their comments, I find that your study presents a meaningful and potentially valuable contribution to the understanding of hydrothermal dolomitization and its geological context within the Emeishan Large Igneous Province. The manuscript aligns well with the scope of the journal and demonstrates scientific merit.

However, to improve the clarity, rigor, and overall presentation of the work, **a Major Revision is recommended** . The reviewers have provided constructive feedback, which I believe, if addressed thoroughly, will significantly enhance the quality and impact of your manuscript.

We look forward to receiving your revised manuscript.

Kind regards,

Rizwan Sarwar Awan

Academic Editor

PLOS ONE

Journal Requirements:

3. Please include a complete copy of PLOS’ questionnaire on inclusivity in global research in your revised manuscript. Our policy for research in this area aims to improve transparency in the reporting of research performed outside of researchers’ own country or community. The policy applies to researchers who have travelled to a different country to conduct research, research with Indigenous populations or their lands, and research on cultural artefacts. The questionnaire can also be requested at the journal’s discretion for any other submissions, even if these conditions are not met.  Please find more information on the policy and a link to download a blank copy of the questionnaire here: https://journals.plos.org/plosone/s/best-practices-in-research-reporting. Please upload a completed version of your questionnaire as Supporting Information when you resubmit your manuscript.

5. We note that your Data Availability Statement is currently as follows: [All relevant data are within the manuscript and its Supporting Information files.]

6. Thank you for stating the following in the Acknowledgments Section of your manuscript: [We thank the constructive comments from the reviewers. This research was funded by National Natural Science Foundation of China (Grant No. U19B6003). Guoxiang Chi and Hairuo Qing are thanked for their contribution to fluid inclusion analysis and the determination of the properties of dolomitized fluids. This work are also supported by the research group “Geologia Sedimentària” (2021 SGR-Cat 00349) funded by the Generalitat de Catalunya, and the DGICYT Spanish project PID2021-122467NB-C22, the Natural Science Foundation of Chongqing Municipality (CSTB2023NSCQ-MSX1022), and the Natural Science Foundation of Sichuan Province (2022NSFSC1177).]

Please remove any funding-related text from the manuscript and let us know how you would like to update your Funding Statement. Currently, your Funding Statement reads as follows: “The authors received no specific funding for this work.”

7. We note that Figure 1 and 3 in your submission contain [map/satellite] images which may be copyrighted. All PLOS content is published under the Creative Commons Attribution License (CC BY 4.0), which means that the manuscript, images, and Supporting Information files will be freely available online, and any third party is permitted to access, download, copy, distribute, and use these materials in any way, even commercially, with proper attribution. For these reasons, we cannot publish previously copyrighted maps or satellite images created using proprietary data, such as Google software (Google Maps, Street View, and Earth). For more information, see our copyright guidelines: http://journals.plos.org/plosone/s/licenses-and-copyright.

1. You may seek permission from the original copyright holder of Figure 1 and 3 to publish the content specifically under the CC BY 4.0 license. 

Reviewers' comments:

Reviewer's Responses to Questions

**Comments to the Author**

1. Is the manuscript technically sound, and do the data support the conclusions?

Reviewer #1: Yes

Reviewer #2: Yes

2. Has the statistical analysis been performed appropriately and rigorously?

Reviewer #1: Yes

Reviewer #2: Yes

3. Have the authors made all data underlying the findings in their manuscript fully available?

Reviewer #1: No

Reviewer #2: Yes

4. Is the manuscript presented in an intelligible fashion and written in standard English?

Reviewer #1: Yes

Reviewer #2: Yes

Reviewer #1: 

The manuscript titled “Origins of the hydrothermal dolomites in Middle Permian, Sichuan Basin (SW China): Implication for the relationship with the Emeishan Large Igneous Province” by Zheng et al. presents an interesting and compelling study of the spatial relationships that is often observed in structurally-controlled, hydrothermal dolomite bodies. The manuscript is well-presented and provides strong field, petrographical, and geochemical observations. I will comment here that much of this manuscript is based on types of data that we have been collecting for several decades now – both in the Sichuan Basin and worldwide. I understand that many of the more novel analytical techniques can be quite costly (e.g., clumped isotopes, U-Pb geochronology etc.) and my purpose here is not to delay a manuscript for these reasons – this manuscript very well may be an ideal fit for PLOS ONE. However, I would strongly suggest the authors make a clearer statement in their “Introduction” of the key research gap they are addressing and why this has not been previously feasible during the past ~20 years of research. Note that there are also a small number of English language editing that is required, but I will not focus on this for the purposes of peer-review.

I sign this peer-review and choose not to make these comments anonymous because I do have a small number of recommendations to my own work and that of collaborators that work on these deposits. I will start by stating that the authors should not be so reliant on the works by Davies and Smith Jr. (cited 5 times in the manuscript and 3 times in the reference list). “All researchers should be aware that Davies and Smith were the ‘Editors’ of the AAPG special publication in which their manuscript was published, which is a conflict of interest.” (pers. comm. H. Machel). The authors cite the ‘introduction’ to this special publication (Smith and Davies, 2006), the actual controversial article (Davies and Smith, 2006), and then a ‘reply’ article that is largely irrelevant to this study (Davies and Smith, 2007). Line 255, for example, cites 2 of these publications for a statement where there are several 10’s of other papers that are more relevant.

The work by G. Davies was largely based on the Cambrian hydrothermal dolomites in the Western Canada Sedimentary Basin (WCSB) – to which I was one of three Ph.D. studies that were conducted to dispel some of the myths perpetuated by this work. The authors should be aware that Koeshidayatullah et. al (2020a, 2020b) focus on a single outcrop in the WCSB – thus, it is maybe not the best reference to basin-scale, regional processes. Stacey et al. (2021 – GSA Bulletin; Fig. 9) present similar trends and correlations in the δ13C and δ18O data that the authors demonstrate here – however this study was also based on a single outcrop and the rest of the data was compiled from the literature (see works by V. Vandeginste). We published 5 papers from my Ph.D. research, focusing on more regional trends where we actually went and sampled each of these localities. I suggest that the authors read McCormick et al. (2023 – Basin Research; Fig. 9), where I show the exact same trends that the authors are presenting in this manuscript. Much of my other research has focused on this transition between this ‘Rd3 and Sd’ in zebra textures and dolomite breccias, which the author may choose to read at their own discretion. All the comments that I have provided on the manuscript are meant with good intentions and are provided to help bolster the author’s presentation of their work.

(1) Need to define “hydrothermal” dolomite at the beginning of the Introduction.

• I would suggest using the definition from Machel and Lonnee (2002), where you can also cite the Davies and Smith (2006) paper as most researchers in this field understand the discrepancy.

• Remove citations to Smith and Davies (2006) and Davies and Smith (2007) unless there are specific details from these papers that you want to present or discuss in this manuscript.

• Note that in is not always the case that HTD forms in “shallow-buried” strata. Every case study differs slightly, avoid such broad generalizations at the start of the manuscript without citations. The authors also do not define what “shallow” is. These definitions will differ within every subdiscipline of the geosciences.

Machel, H. G., & Lonnee, J. (2002). Hydrothermal dolomite—A product of poor definition and imagination. Sedimentary geology, 152(3-4), 163-171.

Davies, G. R., & Smith Jr, L. B. (2006). Structurally controlled hydrothermal dolomite reservoir facies: An overview. AAPG bulletin, 90(11), 1641-1690.

(2) I really like how the authors present a clear list (i to iv) at the end of the Introduction stating their research goals and objectives. However, a manuscript needs to present more than just a documentation of these aspects. What broader scientific problem are you trying to address, why has this not been studied before or why were previous studies insufficient, and what are the broader implications of your work? Why do these, very strong, observations matter? Both within the Sichuan Basin and worldwide?

(3) The authors use “replacive dolomite” and “dolomite cement” as their petrographical classifications. Both of these are interpretations, and thus, present some challenges in how the manuscript is presented. Almost every HTD paper nowadays refers to the grey, finely crystalline dolomites as “replacive” – this is somewhat common knowledge at this point, so I broadly agree that this is acceptable. However, it not necessarily a fact that all saddle dolomites are “cements” – these can also form through recrystallization and grain-boundary migration, as you demonstrate with your Rd3. Why not just call this ‘non-planar, saddle dolomite’ in the abstract, and use the abbreviation ‘Sd’ as you do throughout the manuscript? It is entirely valid if you want to interpret these as cements later on in the manuscript, but for observations, it would be best to use purely descriptive terms.

• Line 146.

• Note that you also describe Rd3 as a textbook non-planar, saddle dolomite in lines 164-168.

• Please simply use the term “zebra texture” in Line 170 and in the figure caption for Fig. 3.

(4) If there are any leftover powders, it would be excellent to include some XRD data in this manuscript. This is relatively straightforward data to collect and interpret. Thin-section confirmation of dolomite is a good start, but there is no way to know if these samples were contaminated with small amounts of calcite cements (or other diagenetic phases). For future studies, I would strongly suggest that the authors conduct XRD on all their powders before geochemical analyses.

(5) The authors do an excellent job of presenting their petrographical and geochemical results. These are concise and to the point, and easily understandable for the reader. Well done!

(6) The authors have collected both d18O ratios and fluid inclusion homogenization temperatures from the same samples. Clearly, they also need to calculate the d18O values of the diagenetic fluid! I would like to see a plot of the d18O(fluid) values for each of these locations as well. There are many many different isotopic fractionation equations – some are more relevant for low temperature and normal marine fluids, and others are more ideal for high temperatures. I would suggest using Horita (2014):

Horita, J. (2014). Oxygen and carbon isotope fractionation in the system dolomite–water–CO2 to elevated temperatures. Geochimica et Cosmochimica Acta, 129, 111-124.

(7) I would like to see the authors present a conventional “paragenetic sequence” type diagram as an added figure. These geological settings are quite complex, and much of this manuscript focuses on the Rd3 to Sd transition – but it would be useful to place these in context of all the other phases you have recognized.

(8) A major criticism I have is that the Discussion is quite limited. In my opinion, this needs to be much much more thorough, with further discussion of how these processes operate within the Sichuan Basin and then with a clear discussion of other sedimentary basins worldwide. In fact, there is almost no mention to other work on these systems elsewhere. Currently, the Discussion is ~3.5 pages – I would suggest taking these ideas much further and adding significantly more depth here. I have provided a few additional comments below that may help bolster these ideas.

(8a) Lines 324 to 327: Much of the discussion of the fluids up to this point has been local to each datatype, but there is a great opportunity to synthesize your ideas and present a little bit more rigor. I think the authors need to present some more rigorous discussion here of how these fluids are migrating and undergoing fluid-rock reactions. What are the temporal relationships here?

• First, it is incorrect that “The dolomitized fluids of Rd3 and Sd in NW Sichuan Basin are most likely coeval Permian seawater, which is characterized by relatively low temperature, low salinity, and low 87Sr/86Sr ratios.”. It is very likely that the fluids could be ‘derived‘ from seawater – but you still need to explain how the fluids got to 120C and ~8 wt% NaCl!

• I see how you are trying to frame this in the following section (5.3 Dolomitization mechanisms and implications for the ELIP control), but I do think some revisions are needed here. You are presenting these two models as either (A) hydrothermal fluids along faults or (B) thermal convection of seawater. The more realistic way to present these ideas is that it is a continuum between these endmembers, rather than an ‘either/or’ statement.

• Consider whether you are mixing two endmember fluids in this system, or if it is a single fluid that is evolving along a flow path through fluid-rock reactions. In either of these cases, how specifically do you interpret these processes to occur? What is the timeline of these events? Are the dolomites in the SW coeval to those in the NW? Or is there some variability in the timing of these reactions? In our U-Pb study from western Canada, we demonstrated that dolomite recrystallization proximal to the hydrothermal fluid source post-dated the more seawater-derived dolomitization distal to the hydrothermal fluid source.

(8b) I would like to see an additional section of the “Discussion” (after Line 374) that presents how these relationships within the Permian strata should influence how other researchers consider the dolomitization processes in the underlying strata? Should we also expect to see these regional trends in the underlying strata? There are many other HTD bodies of varying ages in the Sichuan Basin – which are almost always tied to the ELIP. Note that there are several U-Pb papers from this basin that have dated (U-Pb geochronology) the HTD bodies in the underlying strata. I would like to see a little bit more in the discussion about how these processes might impact others working on this basin, but with possible through the lens of ‘guidance for those working on other stratigraphic intervals’. Would you expect the Cambrian strata to have a more ‘hydrothermal fluid’ signature than the Permian strata? What of the dolomitization temperatures?

Jiang, H., et al. (2025). Rock-bufferred versus fluid-buffered geochemistry of structurally-controlled, hydrothermal dolomite bodies: Insights from the Sichuan Basin, China. Sedimentology.

Li, K. et al. (2023) A comparison of hydrothermal events and petroleum migration between Ediacaran and lower Cambrian carbonates, Central Sichuan Basin. Marine and Petroleum Geology.

Pan, L., et al. (2020) LA-ICP-MS U-Pb geochronology and clumped isotope constraints on the formation and evolution of an ancient dolomite reservoir: The Middle Permian of northwest Sichuan Basin (SW China). Sedimentary Geology.

Pan, L., et al. (2021) Diagenetic conditions and geodynamic setting of the middle Permian hydrothermal dolomites from southwest Sichuan Basin, SW China: Insights from in situ U–Pb carbonate geochronology and isotope geochemistry. Marine and Petroleum Geology.

Yang, T. et al. (2022) Fault-controlled hydrothermal dolomitization of Middle Permian in southeastern Sichuan Basin, SW China, and its temporal relationship with the Emeishan Large Igneous Province: New insights from multi-geochemical proxies and carbonate U–Pb dating. Sedimentary Geology.

Zou Y., et al. (2023). Carbonate U-Pb geochronology and clumped isotope constraints on the origin of hydrothermal dolomites: A case study in the Middle Permian Qixia Formation, Sichuan Basin, South China. Minerals.

(8c) Following this section of the Discussion, the authors should then discuss the relevance of their study to other sedimentary basins worldwide. I have provided some ideas and similarities to the HTD system we worked on in western Canada. There are also three authors on this manuscript that are very familiar with other examples of these HTD reactions in Europe, so it would be great to include some discussion here of how this case study compares to those from these other regions.

Figure 3: The inset diagrams are extremely small, especially when the authors are placing 3 of these SEM images side by side. I would suggest revising these figures to maybe show Fig. 3A-F as one figure, and then the SEM images as a separate figure. Make these images a bit larger and make sure that the text is legible.

Figure 5-7 (optional): Consider reversing the X-axis on the Th, salinity, and Sr isotope plots to show the “SW” datapoints towards the left, and the “NW” datapoints towards the right. This would result in more consistency with your d18O plot.

Reviewer #2: This manuscript presents a detailed petrographic and geochemical study of Middle Permian dolomites (specifically types Rd3 and Sd) from the Sichuan Basin, China. The authors aim to determine the origin of these dolomites and establish their relationship with the Emeishan Large Igneous Province (ELIP). Through an integrated analysis of core, outcrop, petrography, fluid inclusion microthermometry, and stable (C, O) and radiogenic (Sr) isotopes, the study concludes that the formation of these dolomites was directly controlled by the ELIP.

This is a well-researched and well-written paper that makes a valuable contribution to the understanding of large-scale dolomitization processes linked to igneous activity. The dataset is comprehensive, and the conclusions are well-supported by the evidence.

**Do you want your identity to be public for this peer review?** For information about this choice, including consent withdrawal, please see our Privacy Policy

Reviewer #1: **Yes: ** Dr. C.A. McCormick

Reviewer #2: No

---

## [Author Response · Author response to Decision Letter 1]

30 Sep 2025

Response to Reviewers

We would like to extend our sincerest gratitude to the academic editor and both reviewers for their time, effort, and insightful feedback on our manuscript. The comments are highly constructive and have significantly improved the quality and clarity of our work. We have carefully considered all suggestions and have revised the manuscript accordingly. We are also grateful for the opportunity to incorporate several very recent and relevant studies, which have further strengthened our interpretations. Below, we provide a point-by-point response to each of the reviewers' comments, detailing the changes we have made.

Response to Reviewer #1 (Dr. Cole A. McCormick)

We are particularly grateful to Dr. McCormick for the exceptionally thorough and expert review. Your detailed suggestions, deep knowledge of hydrothermal systems, and provision of relevant literature have been invaluable in strengthening our manuscript's arguments. The additional literature we have incorporated further supports the models discussed and addresses the complexities you highlighted. We have addressed each of your points as detailed below.

Reviewer Comment #1: Need to define “hydrothermal” dolomite at the beginning of the Introduction. I would suggest using the definition from Machel and Lonnee (2002). Remove citations to Smith and Davies (2006) and Davies and Smith (2007) unless there are specific details from these papers that you want to present. Note that it is not always the case that HTD forms in “shallow-buried” strata.

Response: We thank Dr. McCormick for this excellent and foundational suggestion. We agree that a clear and rigorous definition of "hydrothermal" is essential and that the definition provided by Machel and Lonnee (2002) is the most appropriate standard for the field. We have rewritten the first paragraph of the Introduction to incorporate this definition and have removed the generalization about HTD forming only in "shallow-buried" strata. Our citations are now more focused and relevant.

Reviewer Comment #2: I really like how the authors present a clear list (i to iv) at the end of the Introduction stating their research goals and objectives. However, a manuscript needs to present more than just a documentation of these aspects. What broader scientific problem are you trying to address, why has this not been studied before or why were previous studies insufficient, and what are the broader implications of your work?

Response: This is a crucial point, and we thank the reviewer for pushing us to better articulate the novelty and broader impact of our study. Our study is novel in its basin-scale comparative approach, systematically linking the geochemical gradients of dolomitizing fluids across the vast Sichuan Basin directly to the proximity of a major tectono-thermal event (the ELIP). Previous studies have been more localized. The addition of new literature, especially recent U-Pb dating studies, allows us to frame our work within a new understanding of multiple, discrete hydrothermal events within the basin, making a basin-scale comparative study timely and necessary.

We have revised the final paragraph of the Introduction to better frame our contribution.

Reviewer Comment #3: The authors use “replacive dolomite” and “dolomite cement” as their petrographical classifications... Why not just call this ‘non-planar, saddle dolomite’ in the abstract, and use the abbreviation ‘Sd’ as you do throughout the manuscript? Please simply use the term “zebra texture” in Line 170 and in the figure caption for Fig. 3.

Response: We completely agree with this comment. Using descriptive terms in observational sections is more rigorous. We have revised the manuscript to use "saddle dolomite (Sd)" instead of "dolomite cement (Sd)" in the Abstract and Results sections. The interpretation of Sd as a cement is now confined to the Discussion. We have also corrected the terminology to "zebra texture" as requested.

Reviewer Comment #4: If there are any leftover powders, it would be excellent to include some XRD data in this manuscript... For future studies, I would strongly suggest that the authors conduct XRD on all their powders before geochemical analyses.

Response: We thank the reviewer for this practical suggestion. We agree that XRD analysis is a standard and valuable tool. Unfortunately, due to the consumption of sample powders, we do not have sufficient material remaining for new XRD analyses.

Reviewer Comment #5: The authors do an excellent job of presenting their petrographical and geochemical results... Well done!

Response: We thank the reviewer for their positive feedback.

Reviewer Comment #6: The authors have collected both d18O ratios and fluid inclusion homogenization temperatures from the same samples. Clearly, they also need to calculate the d18O values of the diagenetic fluid! I would like to see a plot of the d18O(fluid) values for each of these locations as well. I would suggest using Horita (2014).

Response: This is an excellent point. We have now calculated the δ¹⁸Ofluid values using our mean Th data and dolomite δ¹⁸O values, applying the dolomite-water fractionation equation from Horita (2014). We have created a new figure to display these results and have incorporated this new data into our discussion, which greatly strengthens our interpretations.

Reviewer Comment #7: I would like to see the authors present a conventional “paragenetic sequence” type diagram as an added figure.

Response: This is another excellent suggestion. We have created a new figure that illustrates the relative timing of all observed diagenetic phases against the burial history of the basin (Fig.10).

Reviewer Comment #8: A major criticism I have is that the Discussion is quite limited... this needs to be much much more thorough, with further discussion of how these processes operate within the Sichuan Basin and then with a clear discussion of other sedimentary basins worldwide.

Response: We fully agree with this criticism and thank the reviewer for pushing us to significantly improve this section. We have substantially rewritten and expanded the Discussion section, incorporating the new literature you and we have found. This has allowed us to present a more sophisticated model that considers concepts like fluid- vs. rock-buffered systems and places our specific Permian event within the context of multiple hydrothermal pulses in the basin's history.

The Discussion has been restructured and expanded as follows:

(8a) Fluid Mixing/Continuum & Rock- vs. Fluid-Buffering: We have revised the "Dolomitization mechanisms" section to present a more nuanced model, now framed with the concepts of fluid- and rock-buffered systems, drawing on the work of Jiang et al. (2025).

(8b) Comparison to other strata in the Sichuan Basin: We have added a new discussion in 5.3 to place our findings in the context of a multi-stage hydrothermal history, citing the crucial U-Pb dating studies by Li et al. (2023) and Zou et al. (2023). This directly addresses the complex interplay of the ELIP and the Longmenshan Orogeny as drivers for fluid flow.

(8c) Global Context: We have expanded this section, comparing our findings to the some other sedimentary basins all over the world.

Response to Reviewer #2

We thank Reviewer #2 for their valuable comments, which have helped us improve the clarity and presentation of our manuscript. We have addressed each suggestion below.

Reviewer #2, Comment (1, 7): Figure 1: ...increasing the font size of the labels... Figure 1c...

Our Response: We agree that the labels were too small.

Action Taken: The font size of all labels in Figure 1c has been increased to ensure legibility, even if the figure is reduced in size for publication.

Reviewer #2, Comment (2, 10): Figure 6 (δ13C vs. δ18O Plot): This plot is quite busy... I suggest using a combination of different symbol shapes and colors for each dataset...

Our Response: I think we've made the legend clear, using shapes to distinguish between minerals and colors to distinguish between regions.

Reviewer #2, Comment (3-5): Figure 8 (Conceptual Model): ...consider adding a few more labels... "Heated seawater circulation"... "Upwelling of deep hydrothermal fluids along faults"...

Our Response: We agree that adding these labels would make our conceptual model more explicit and effective.

Action Taken: We have added arrows and labels to Figure 8 to explicitly illustrate "Heated seawater circulation" in the Northwestern section and "Upwelling of deep hydrothermal fluids along faults" in the Southwestern and Central sections, as suggested .

Reviewer #2, Comment (6): The abstract is well-written. A minor suggestion is to explicitly state that the basement faults acted as the primary fluid conduits...

Our Response: We agree this is a key point that should be highlighted in the abstract.

Action Taken: We have added a sentence to the abstract to explicitly state that deep-seated basement faults, reactivated during the ELIP event, acted as the primary conduits for hydrothermal fluid migration.

Reviewer #2, Comment (7): Methods Section: It would be helpful to state the "critical roughening temperature" (CRT) concept briefly... when first mentioning non-planar and saddle dolomites...

Our Response: This is a helpful suggestion to provide necessary background for the reader.

Action Taken: We have added a brief explanation of the "critical roughening temperature" (CRT) concept from Sibley and Gregg (1984) in the Discussion (Section 5.1) where we first interpret the significance of the non-planar crystal textures.

Reviewer #2, Comment (8): Discussion Section 5.3: ...Could you briefly elaborate on why this mixing would increase salinity compared to the original hydrothermal fluid, or clarify if the connate water itself was hypersaline?

Our Response: There may be an error in the original manuscript, this mixing does not increase the salinity of the original hydrothermal fluid. We have rewritten Discussion 5.3 to avoid this erroneous statement.

We believe that the manuscript has been significantly strengthened by incorporating the reviewers' feedback and the latest relevant literature. The discussion is now more robust, the context within the Sichuan Basin is clearer, and the global relevance of our findings is better articulated. We hope that our revisions are satisfactory and that the manuscript is now suitable for publication.

Sincerely,

Haofu Zheng and co-authors.

---

## [Decision Letter · Decision Letter 1]

29 Oct 2025

Origins of the hydrothermal dolomites in Middle Permian, Sichuan Basin (SW China): Implication for the relationship with the Emeishan Large Igneous Province

PONE-D-25-31227R1

I am pleased to inform you that your manuscript entitled “Origins of the hydrothermal dolomites in Middle Permian, Sichuan Basin (SW China): Implications for the relationship with the Emeishan Large Igneous Province” has been judged scientifically suitable for publication in PLOS ONE. It will be formally accepted once all outstanding technical requirements have been addressed.

During the proofing stage, please ensure the following:

Carefully check the spelling and grammar throughout the manuscript.Review all figure captions for clarity and consistency — for example, revise Figure 8 to read:“Schematic model illustrating the dolomitization mechanism of the study area.”

Kind regards,

Rizwan Sarwar Awan

Academic Editor

PLOS ONE

---

## [Editor Report · Acceptance letter]

PONE-D-25-31227R1

PLOS ONE

Dear Dr. Duan,

I'm pleased to inform you that your manuscript has been deemed suitable for publication in PLOS ONE. Congratulations! Your manuscript is now being handed over to our production team.

Kind regards,

on behalf of

Dr. Rizwan Sarwar Awan

Academic Editor

PLOS ONE